# Glide-snow avalanches: A mechanical, threshold-based release area model

Amelie Fees[1], Alec van Herwijnen[1], Michael Lombardo[1], Jürg Schweizer[1], and Peter Lehmann[2]

[1]WSL-Institute for Snow and Avalanche Research SLF, Davos, Schweiz
[2]Physics of Soils and Terrestrial Ecosystems, ETH Zürich, Switzerland

**Correspondence:** Amelie Fees (amelie.fees@slf.ch)

**Abstract.**

Glide-snow avalanches release at the ground-snow interface due to a loss in basal friction. They pose a threat to infrastructure because of the combination of unreliable mitigation measures, limited forecasting capabilities, and a lack of understanding of the release process. The aim of this study was to investigate the influence of spatial variability in basal friction and snowpack properties on the avalanche release area distribution and the release location. We developed a pseudo-3D, mechanical, threshold-based model that consists of many interacting snow columns on a uniform slope. Parameterizations in the model are based on our current understanding of glide-snow avalanche release. The model can reproduce the power law glide-snow avalanche release area distribution as observed on Dorfberg (Davos, Switzerland). A sensitivity analysis of the input parameters showed that the avalanche release area distribution was mostly influenced by the homogeneity (correlation length and variance) of the basal friction and whether the basal friction was reduced suddenly or in small increments. Larger release areas were modeled for a sudden decrease and a more homogeneous basal friction. The spatial variability of the snowpack parameters had little influence on the release area distribution. Extending the model to a real-world slope showed that the modeled location of avalanche releases qualitatively matched the observed locations. The model can help narrow down the length- and time-scales for field investigations. Simultaneously, it can grow in complexity with our increasing knowledge on glide-snow avalanche release processes. Input parameters such as the basal friction or snowpack parameters could potentially all be connected to the liquid water content. This would allow for the use of meteorological measurements to drive the model. The model has the potential to help identify potentially dangerous conditions for large or numerous avalanches which would help improve glide-snow avalanche forecasting.

# 1 Introduction

Glide-snow avalanches fail at the ground-snow interface, which can result in the release of large snow volumes that endanger infrastructure in mountain regions (Clarke and McClung, 1999; Mitterer and Schweizer, 2012; Peitzsch et al., 2015). These avalanches pose a threat that is difficult to mitigate due to limited forecasting capabilities (Simenhois and Birkeland, 2010; Jones, 2004) and unreliable mitigation measures (Sharaf et al., 2008; Jones, 2004). Observations have shown that glide-snow avalanches mostly release in well-known avalanche paths which are typically characterized by a slope angle greater than 28°

(Ancey and Bain, 2015) and a smooth ground surface (in der Gand and Zupančič, 1966). It is generally accepted that the loss of friction between the snowpack and the ground is caused by liquid water at the ground-snow interface (Clarke and McClung, 1999; in der Gand and Zupančič, 1966; McClung, 1987).

The potential sources of liquid water include melt-water percolation (Lackinger, 1987; Clarke and McClung, 1999), geothermal heat (McClung, 1987; Newesely et al., 2000; Höller, 2001), and capillary suction (Mitterer and Schweizer, 2012). Whether

the loss in friction causes the formation of a tensile crack or a full-depth avalanche release depends on the stauchwall, which is the supporting snow cover located at the lower edge of the gliding zone. The stauchwall stabilizes the gliding snowpack as long as it can withstand the increased rate of loading after tensile failure (Bartelt et al., 2012). There have been several attempts to analytically describe the glide velocity (Haefeli, 1939; McClung, 1981, 1987; Bombelli et al., 2021) and release through stauchwall failure (Bartelt et al., 2012). However, none of these models investigate a pseudo-3D slope or the effect of soil and

snow spatial variability on avalanche size, location, and release timing.

While the processes leading to glide-snow avalanche release are not fully understood, we can think of glide-snow avalanches as a gravity-driven mass movement. It has been shown for other mass movements such as dry-snow slab avalanches (Kronholm and Birkeland, 2005; Faillettaz et al., 2004), landslides (Lehmann and Or, 2012; Stark and Hovius, 2001), and rockfalls (Dussauge et al., 2003; Malamud et al., 2004) that their release area distributions follow a scale-invariant power law distribution.

This means that the probability distribution $p(x)$ of release areas follows a power law with an exponent $\alpha$ (Bak, 1996; Sornette, 2006) corresponding to:

$$p(x) \propto \left( \frac{x}{x_{\min}} \right)^{-\alpha} \text{ with } x > x_{\min}, \alpha > 1. \tag{1}$$

Often, the power law only applies to the tail of a distribution where values are greater than a minimum value ($x_{\min}$) (Sornette, 2006). These heavy-tailed power law distributions may be associated with self-organized criticality (SOC), which refers to the

spontaneous organization of an externally driven system into a (marginally) stable state. Models that replicate this behaviour are called cellular automata. They consist of many interacting elements that show a non-linear threshold response while externally driven with a constant rate (e.g. Sornette, 2006). In other words, local element failures can progress into large-scale mass release.

A range of cellular automata have been introduced including, but not limited to, the sandpile cellular automaton, spring-

block-models or fibre bundle models. The sandpile cellular automaton (BTW model, Bak et al., 1987; Hergarten, 2002; Piegari et al., 2006) is built on a threshold-based instantaneous mass redistribution among the neighboring elements. Spring-Block-Models such as the Burridge-Knopoff model (Burridge and Knopoff, 1967) or the OFC-model (Olami et al., 1992) take into

account mechanical properties between neighboring elements. They have previously been extended to account for friction, sliding and tension cracking in gravity-driven systems (Faillettaz et al., 2010). Fibre bundle models (Alava et al., 2006; Lehmann and Or, 2012) have been used to study fracture processes including dry-snow failure (Reiweger et al., 2009; Capelli et al., 2018). These SOC concepts have been applied successfully to model gravity-driven mass movements including landslides (Lehmann and Or, 2012) and dry-snow slab avalanches (Kronholm and Birkeland, 2005; Faillettaz et al., 2004).

In this study, we show that the same SOC concepts can be applied for glide-snow avalanches. We show (i) that the frequency distribution of glide-snow avalanche release areas observed on Dorfberg can be described with a power law and (ii) that the power law exponent can be reproduced with a threshold-based mechanical model that was based on the principles of SOC. The mechanical interaction and failure propagation between elements (snow columns) was parameterized according to the current process understanding of glide-snow avalanche release. The aim of this study was to investigate the influence of spatial variations in basal friction and snow properties on the avalanche release area and the power law exponent.

## 2   Release area model

The glide-snow avalanche release area model was inspired by the landslide triggering model described in Lehmann and Or (2012). To model the avalanche release area based on the progression of local failures, we discretized the snowpack into interacting snow columns connected to the soil through the basal friction. In contrast to the landslide model, which accounts for reduced soil strength by infiltrating water, the model presented here is driven by a uniform, step-wise reduction in basal friction. The reduction in basal friction can lead to locally unstable snow columns and stress redistribution onto neighboring columns which can further lead to failure propagation and avalanche release. The basal friction is a proxy and we do not include any assumptions on how the basal friction is linked to processes such as liquid water formation or environmental variables such as ground roughness. In this section, we first describe the model implementation and parameterizations on a uniform slope. In subsection 2.5 we describe the model adjustments for simulations on a non-uniform slope (real topography) in detail.

### 2.1   Model setup and initiation

The snowpack is discretized into hexagonal columns (apothem $a$, snow height $h$, snow density $\rho$, see Figure 1). The column height is parallel to the direction of gravity and the columns are placed on a uniform hillslope with a slope angle $\beta$ (for implementation of local topography see subsection 2.5). The basal friction ($\mu \in [0,1]$) between the hillslope (ground) and the snow columns was modelled as a Gaussian random field with an exponential covariance function (parameters: variance, mean and correlation length; see Appendix A). The total mass ($m$) of a snow column and its force ($F_G = mg$) due to gravity ($g$) can be divided into two components: the down-slope force ($F_H = F_G \sin\beta$) and the counteracting normal frictional force ($F_N = \mu F_G \cos\beta$). Dividing the forces with the effective hexagonal cross section ($A_H = 2\sqrt{3}a$) results in the normal stress

$$\sigma_N = \frac{F_N \cos\beta}{A_H} = \mu\rho gh\cos^2\beta \tag{2}$$

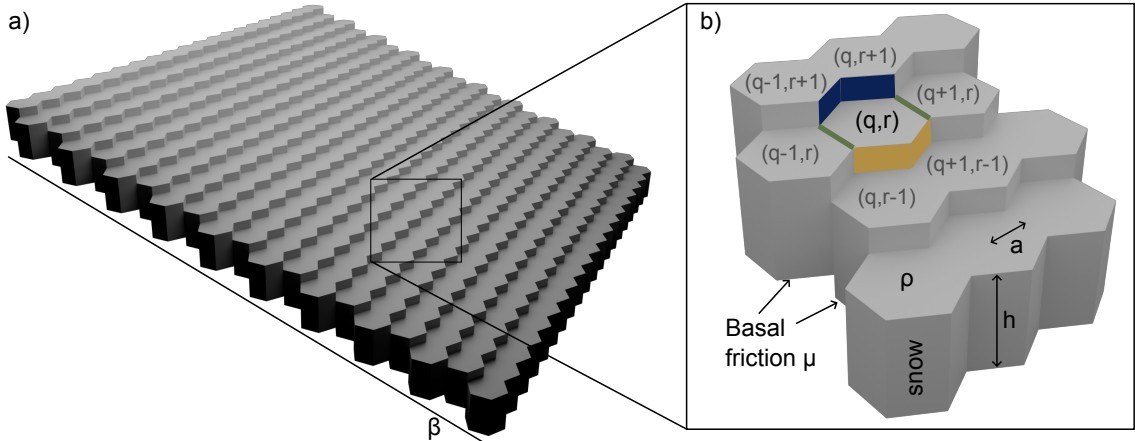

**Figure 1.** a) The model consists of hexagonal columns on a uniform slope $\beta$. b) Every snow column is defined by its snow height $h$, the apothem $a$, the snow density $\rho$ and the basal friction $\mu$. Every column (axial coordinate: (q,r)) is connected to its neighbor through a compressive (yellow), shear (green) or tensile (blue) bond.

and the basal shear stress

$$\tau = \frac{F_\text{H}\cos\beta}{A_\text{H}} = \rho g h \sin\beta\cos\beta. \tag{3}$$

The 'driving' component of the model is the excess stress $W = \sigma_\text{N} - \tau$. During model initialization (Figure 2), the random basal friction field is scaled such that all hexagonal columns are initially stable ($W \geq 0$). The reduction in basal friction due to liquid water formation is mimicked as a step-wise and uniform reduction in basal friction. If this causes the basal shear stress to exceed the normal stress ($W < 0$) anywhere in the model domain, the corresponding column's basal friction is set to the residual friction ($\mu_\text{res} << \mu$) and the excess stress $W$ is recalculated. The residual friction $\mu_\text{res}$ is a constant value which is
independent of the column location and the initial friction $\mu$.

## 2.2 Bonds representing mechanical interaction between snow columns

If a column with the axial coordinate (q,r) is unstable (W<0), it can be stabilized if the excess stress $W$ can be equalized by the strength of the connections to its neighboring columns (Figure 1b). The connection of column (q,r) to its neighbors is referred to as a 'bond' and is a snow strength property. We distinguish between compressive ($f_c$), shear ($f_s$), and tensile ($f_t$) bonds. For a uniform slope, the down-slope neighbors ((q, r-1), (q+1, r-1)) are connected through compressive bonds, the up-slope neighbors ((q-1, r+1), (q, r+1)) through tensile bonds, and the left (q-1, r) and right (q+1, r) neighbors through shear bonds (Figure 1b). The stress is divided equally between bonds of the same type. Subsection 2.5 describes how the bonds are assigned and how the excess stress is distributed in case of a complex topography.

## 2.3 Compressive bond strength and snow density

The strength of the compressive bond ($f_c$) is a snow strength property that was calculated based on the dry-snow density $\rho$. We model the snow density as a Gaussian random field with an exponential covariance function. The density random field is not correlated with the basal friction random field. The spatial variation in density includes all contributions to the column mass as we assume a uniform snow height.

To link the snow density and the compressive bond strength, we followed the dry snow strength ($\sigma_M$)-density relationship reported by Mellor (1975, Figure 17) which we parameterized as

$$\sigma_M = A \exp(B\rho) \text{ for } \rho \leq 500 \text{ kg m}^{-3}. \tag{4}$$

with $A = 256$ Pa and $B = 16.5 \times 10^{-3} \text{m}^3\text{kg}^{-1}$. The snow strength $\sigma_M$ is given in Pa and the snow density $\rho$ in kg m$^{-3}$. As the side length of the hexagonal column influences the area that connects neighboring columns (and thus the acting stress), we introduced a unitless factor $s_\text{strength}$ that allows us to scale the bond strength

$$f_c = s_\text{strength}\sigma_M. \tag{5}$$

The outermost columns that do not have a neighbor are assigned a virtual neighbor with a high predefined compressive bond strength (Table 1), which results in a fixed boundary condition. For the relative magnitude of compressive, tensile, and shear bond strengths ($f_c$:$f_s$:$f_t$), we assume a ratio of 10:2:1 based on the review by Mellor (1975).

## 2.4 Stress redistribution and stability evaluation

The model is initialized with all columns in a stable state ($W \geq 0$). The basal friction is reduced step-wise and uniformly for all columns until an unstable column ($W < 0$) occurs in the system. An unstable column (q,r) initially distributes its excess stress $W = \mu_\text{res}\rho gh \cos^2 \beta - \tau$ equally onto all neighbor bonds. The stress acting on a bond is calculated based on the side wall area of the hexagon. If the stress onto a neighbor (compressive $\sigma_c$, shear $\tau_s$, or tensile $\sigma_t$) exceeds the bond strength (for example: $\sigma_c > f_c$), the bond fails and the stress is redistributed equally amongst the remaining intact bonds. This can lead to stabilization of

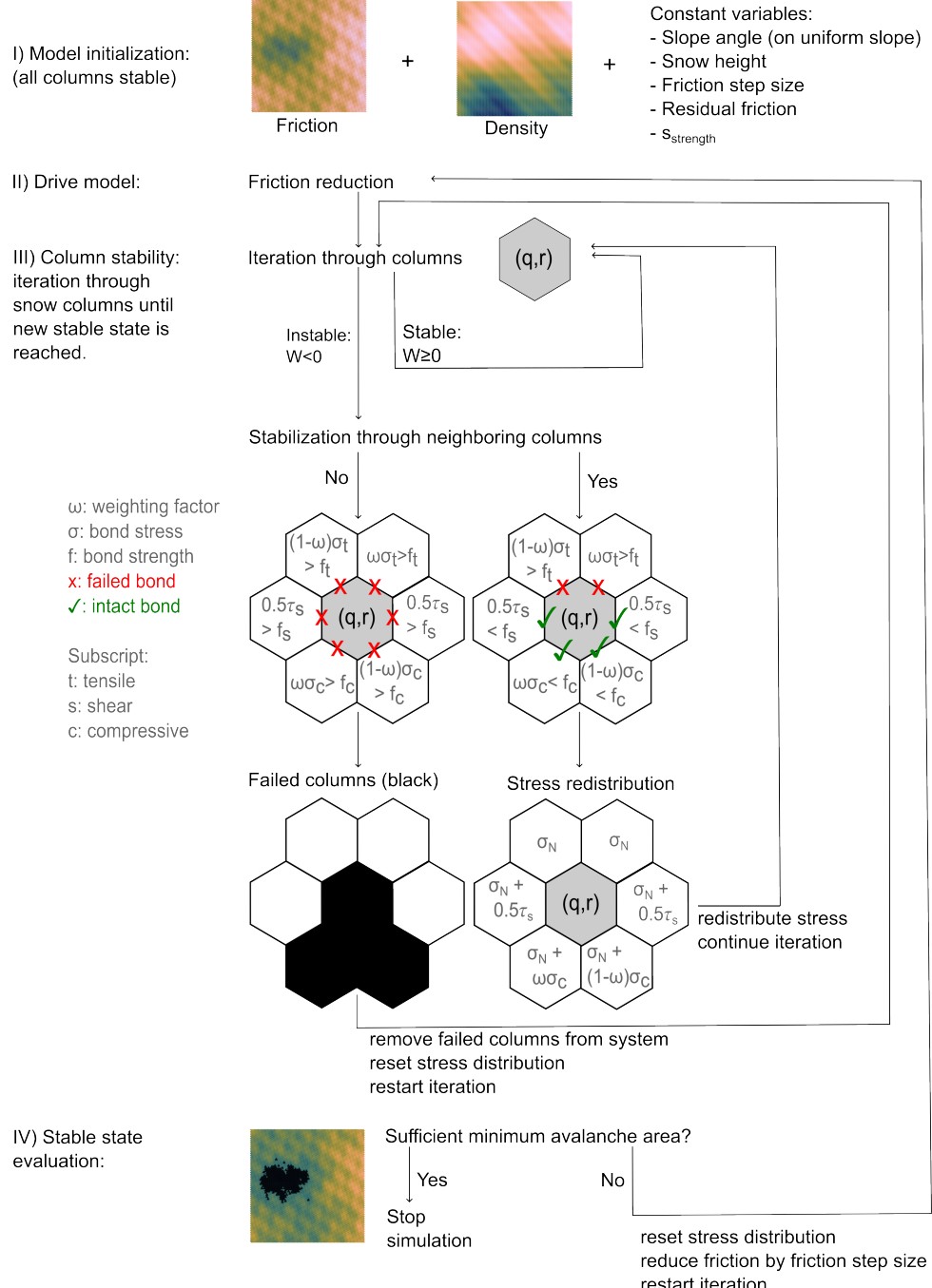

**Figure 2.** Conceptual overview of the modeling steps. After model initialization (subsection 2.1) column stability is evaluated for every column (subsection 2.4). Columns that failed (no stabilization, here indicated by black hexagons) are removed from the system and the system is reevaluated until no additional columns fail.

the column (q,r) if the bonds are strong enough or failure of the column if all bond strengths were exceeded. If a column (q,r) can be stabilized the stress is transferred along the intact bonds onto the neighboring columns. No stress is transferred along a broken bond and no memory of the broken bond is kept for the next neighbor stability evaluation. The transferred stresses are taken into account when recalculating the excess stress corresponding to

$$W = \sigma_N + \sigma_c + \tau_s + \sigma_t - \tau. \tag{6}$$

This redistribution can result in high, local stresses that can cause initially stable columns to overcome frictional support and fail.

If, for column (q,r), all neighboring bond strengths were exceeded, the column (q,r) fails and is removed from the system. In addition, its two compressive neighbors fail as they were unable to support the column (Figure 2). This mimics the partial failure of the stauchwall. If a column failure occurs, the column(s) are removed from the system and they cannot support neighboring columns any longer. This can cause a cascading chain reaction that results in an avalanche of failures.

After the failed columns have been removed, the load redistribution (which occurred during column stabilization) is reset for the entire system and the stability of all remaining unstable ($W < 0$) columns is reevaluated. The system reaches a new stable state when all unstable columns failed or were stabilized by their neighbors. In this stable state only a reduction in friction can cause new failures. The model stops once at least one of the avalanche(s) (clusters of failed columns) in the new stable state exceeds the given minimum avalanche size. If no avalanche exceeds the minimum avalanche size, the basal friction is reduced uniformly by a value defined as the friction step size and the stability evaluation for all columns is restarted. The friction step size acts as a pseudo time variable. A large friction step size results in a large and sudden reduction of basal friction. A small friction step size results in a smaller and more gradual reduction of basal friction.

## 2.5   Model applied to a non-uniform slope

The assumption of a uniform slope simplifies the model because parameters such as the slope angle or the relative coordinates of the compressive, shear and tensile bonds are constant. To run the model on a real slope with varying topography, based on its digital elevation model (DEM), a few adjustments have to be made. In the initialization phase (subsection 2.1, Figure 2) the down-slope gradient and slope angle were calculated at the DEM resolution. To transfer these values from the raster DEM to the hexagonal grid, the values were interpolated to the hexagon center positions. We used nearest-neighbor interpolation for the gradient and linear interpolation for the slope angle.

In the stability evaluation phase (subsection 2.4, Figure 2), the bond type for every neighboring column depends on its relative location to the column (q,r) and the down-slope gradient (Figure 3). Of the six column neighbors, the two down-slope neighbors interact through compressive bonds, the two up-slope neighbors through tensile bonds, and the remaining two neighbors through shear bonds. The direction of the down-slope gradient determines what neighbors are assigned the compressive bonds and how the total compressive stress is weighed among the two compressive neighbors. Figure 3a shows an example where the down-slope gradient direction is given by the angle $\gamma$. Here, the neighbors (0,1) and (1,0) are compressive

neighbors. The weighting factor

$$\omega(\gamma) = \frac{1}{2}\cos 3\gamma + \frac{1}{2} \tag{7}$$

is defined by a cosine function. The neighbors would share the load equally ($\omega = 0.5$) if $\gamma = 30°$ and the compressive strength of neighbor (1,0) would have to carry all of the compressive stress if $\gamma = 0°$. Once the compressive neighbors were determined, the tensile neighbors (here: (-1, 0), (0,-1)) are determined along the same axis (axial coordinates). The weighting factor is the same as for the compressive neighbor along the same axial axis. Finally, the shear neighbors (here: (-1,1), (1,-1)) always share

the load equally ($\omega = 0.5$).

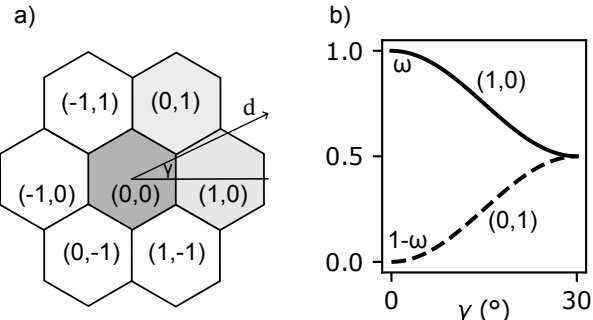

**Figure 3.** Visualization of the stress distribution from the center hexagon (0,0) onto the compressive bonds to neighbors (1,0) and (0,1) on a non-uniform topography. a) The down-slope gradient $\boldsymbol{d}$ and its direction $\gamma$ were calculated from the DEM and b) the parametrization of the stress weighting factor $\omega$ with the gradient direction $\gamma$ for the compressive neighbors (1,0) and (0,1) (Equation 7).

## 3 Sensitivity analysis

### 3.1 Sensitivity analysis on uniform slope

We investigated the influence of snow and friction related model input parameters (Table 1) on the release area distribution and compared the results to the release area distribution observed at Dorfberg (Figure 4a). Dorfberg is a mostly southeast-facing

slope above Davos, Switzerland with well documented glide-snow avalanche activity (Fees et al., 2023). The distribution of glide-snow avalanche release areas (> 10 m$^2$, n = 488) were obtained from georeferenced time-lapse photographs that were taken during the seasons of 2008/09 to 2022/23 (for method and more information see Fees et al. (2023)).

The snow and friction related model input parameters were varied one parameter at a time based on the baseline simulation (Table 1). For one set of input parameters we conducted 30 simulations which stopped once an avalanche in a stable state

exceeded the minimum avalanche size of 10 m$^2$. All simulations were conducted on a uniform slope consisting of 100 x 100 hexagonal columns with a cross section of 1 m$^2$. Details on the model parameters such as the number of simulations, the exponential covariance function, and the system size are provided in Appendix A. The random fields were generated with

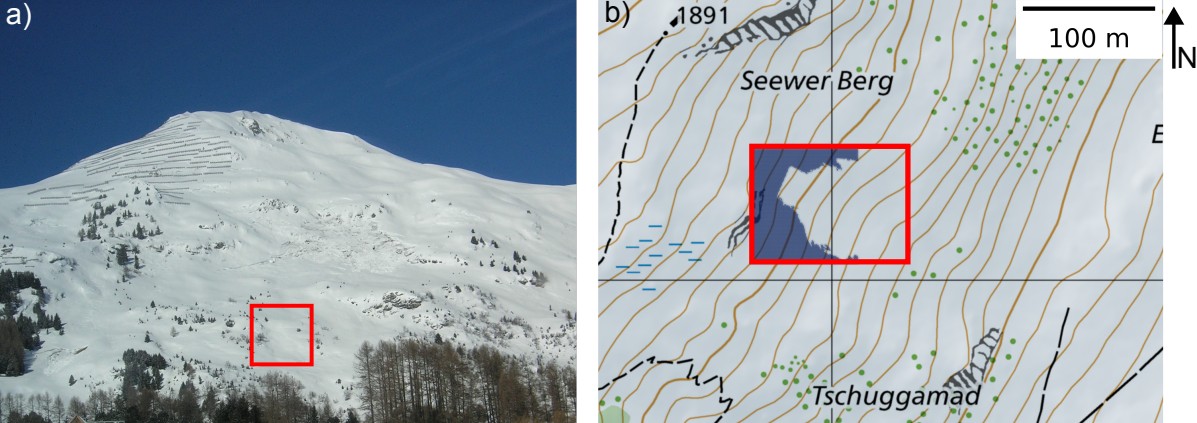

**Figure 4.** a) Overview of the Dorfberg field site (Davos, Switzerland) with the Seewer Berg slope (red box) which was used to evaluate the model on a real topography. b) Map of the area around the Seewer Berg slope (red box). The blue shaded area in the red box indicates the area that was masked out for the model due to high vegetation roughness. Map: Federal Office of Topography

GSTools (Müller et al., 2022). In case more than one avalanche released during a model run, all avalanches above the minimum avalanche size were taken into account for the release area distribution. The power law exponent was estimated with the

175 maximum likelihood estimate. The minimum value $x_{\min}$ was determined according to Clauset et al. (2009) (using Alstott et al. (2014) with the Euclidean metric). If indicated $x_{\min}$ of a simulated distribution was set to the $x_{\min}$ observed on Dorfberg for better comparability between distributions. The aspect ratio ($\frac{\text{width}}{\text{length}}$) of the modelled release area was calculated as the distance between the leftmost and rightmost hexagon (width) and the uppermost and lowermost hexagon (length).

## 3.2 Real topography

We also investigated the influence of a real topography on the release area distribution and the release location. We ran the simulations on the DEM of one slope called Seewer Berg on Dorfberg that is known to produce glide-snow avalanches (Figure 4, coordinates: 46.8183° N, 9.8367° E). The Seewer Berg slope ranges in elevation from 1765 m a.s.l. to 1818 m a.s.l and is southeast-facing with a slope angle of $31° \pm 5°$ (mean $\pm$ standard deviation). There are locations with slope angles around 40°

within the Seewer Berg slope which are interspersed with rocks, trees or large bushes. There have only been few observations of glide-snow avalanches releasing in these locations. As our model does not take vegetation or surface roughness into account, we masked out these parts of the slope to prevent unrealistic avalanche releases in these steep areas (dark blue shaded area in Figure 4b). In the model, the hexagonal cross section was set equal to the DEM resolution (0.25 m$^2$) resulting in a hexagon apothem $a$ of 0.072 m.

**Table 1.** Initialization parameters for the baseline simulation and the sensitivity analysis (both on a uniform slope) and on the Seewer Berg slope topography. The constant boundary conditions are discussed in more detail in Appendix A. The slope angle mean and standard deviation are given for the Seewer Berg topography.

| | Parameters | Baseline | Sensitivity analysis | Seewer Berg |
|---|---|---|---|---|
| | Number of hexagons | 100 x 100 | 100 x 100 | 219 x 187 |
| | Number of simulation runs | 30 | 30 | 30 |
| Constant | Hexagon apothem $a$ (m) | 0.54 | 0.54 | 0.072 |
| | Boundary condition: $f_c$ (Pa) | $10^{50}$ | $10^{50}$ | $10^{50}$ |
| | Ratio: $f_c : f_s : f_t$ | 10:2:1 | 10:2:1 | 10:2:1 |
| | Minimum avalanche area (m$^2$) | 10 | 10 | 10 |
| | Slope angle $\beta$ (°) | 35 | 27 - 40 | $31 \pm 5$ (DEM) |
| Basal | Friction variance | $10^{-4}$ | $10^{-6} - 1$ | $10^{-3}$ |
| friction | Friction correlation length (m) | 350 | 50-500 | 10 |
| | Friction step size | $5 \times 10^{-3}$ | $10^{-6} - 10^{-2}$ | 0.15 |
| | Residual friction $\mu_{\text{res}}$ | 0.01 | 0 - 0.1 | 0.01 |
| | Snow height $h$ (m) | 1 | 0.5 - 10 | 1 |
| | Density $\rho$ (kg m$^{-3}$) | 250 | 150 - 450 | 250 |
| Snow | Density variance | 0 | 0 - 100 | 0 |
| | Density correlation length (m) | 0 | 0 - 500 | 0 |
| | $s_{\text{strength}}$ | $6.25 \times 10^{-3}$ | $3 \times 10^{-3} - 3 \times 10^{-2}$ | $6.25 \times 10^{-3}$ |

## 4 Results

### 4.1 Release area distribution

The frequency distribution of glide-snow avalanche release areas observed on Dorfberg can be described with a power law. This results in a power law exponent $\alpha_{\text{Dorfberg}} = 2.4 \pm 0.1$ and a minimum value $x_{\min} = 633$ m$^2$ (Figure 5). The mean aspect ratio of the release area distribution was $\frac{\text{width}}{\text{length}} = 1.7 \pm 0.7$. The baseline simulation (Table 1) resulted in a power law exponent of $\alpha = 2.5 \pm 0.3$ (using $x_{\min} = 633$ m$^2$, Figure 5) which is within the range of uncertainty of the Dorfberg field data. Modelled avalanches were more often longer than wide, which resulted in a lower aspect ratio of $\frac{\text{width}}{\text{length}} = 0.8 \pm 0.3$.

### 4.2 Sensitivity analysis: basal friction and snow cover

The input parameters (Table 1) related to the basal friction (correlation length, variance and friction reduction step size) substantially influenced the release area distribution and its power law exponent $\alpha$. An increase in basal friction correlation length and a decrease in variance resulted in a more homogeneous basal friction random field. The more homogeneous basal friction resulted in overall more and larger avalanche release areas (Figure 6). The simulation was highly sensitive to the basal friction

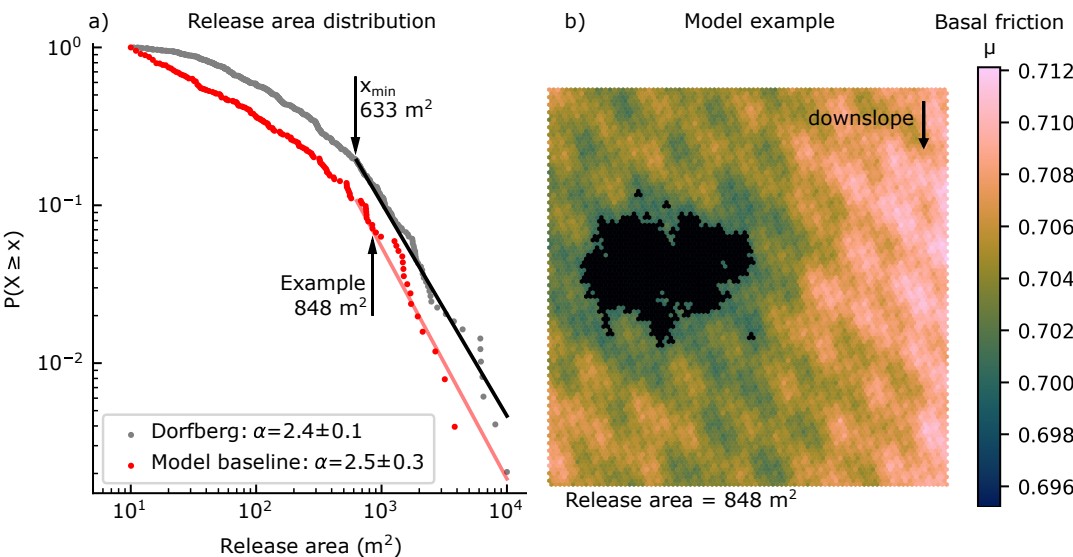

**Figure 5.** a) Complementary cumulative distribution function $P(X \geq x) \propto x^{-(\alpha-1)}$ for glide-snow avalanche release areas on Dorfberg (gray dots) and for the baseline simulation (red dots, Table 1). The corresponding lines indicate the power law fit. b) Example of a large modeled release area that was initiated with the baseline parameters (arrow in a)); in a typical model run additional small release areas ($> 10$ m$^2$) were observed around the main release area.

variance. Small variances ($\sim 10^{-6}$) resulted in an almost uniform friction distribution which caused the entire slope to release as one large avalanche. Large variances ($\sim 1$) resulted in a highly inhomogenous basal friction distribution which caused only small avalanches to release. The power law exponent $\alpha$ was influenced by both the correlation length and the variance, but did not show a clear trend (Figure 6). Comparable release areas and the power law exponent for the Dorfberg field site were obtained for a correlation length of 350 m and a variance of $10^{-4}$. As long as the minimum avalanche size was not reached in a model run, the basal friction was reduced uniformly in increments given by the friction step size. Larger step sizes resulted in more and larger avalanche release areas. The power law exponent $\alpha$ decreased between a step size of 0.002 and 0.006 and subsequently increased with step size. The field observations were reproduced with a step size of 0.005 (Figure 6c).

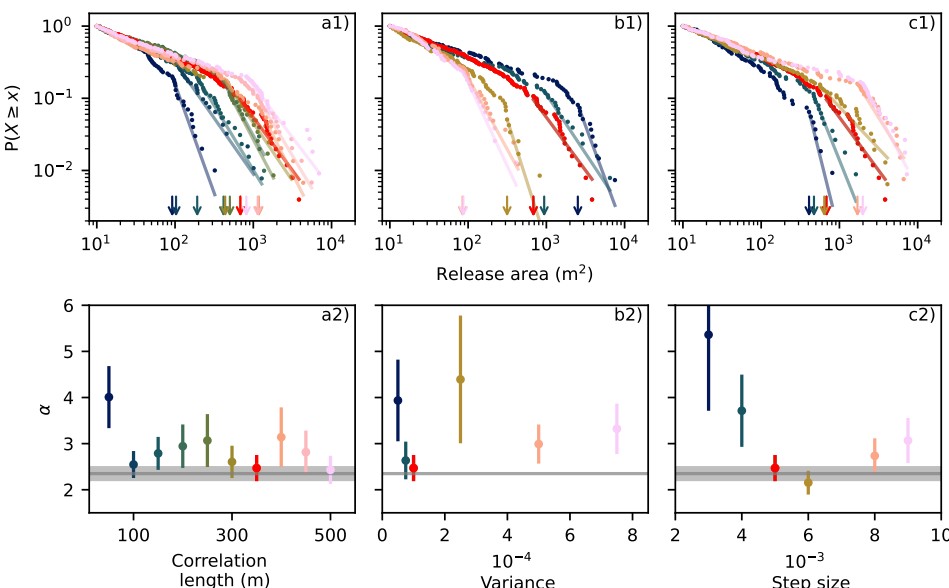

**Figure 6.** Effect of the basal friction on the release area distribution and its power law fit for a range of: a1) basal friction correlation lengths, b1) basal friction variances and c1) step sizes in basal friction reduction. The color indicates the parameter value as given in the graph below. The arrows indicate $x_{min}$ which was determined according to Clauset et al. (2009). The power law exponent $\alpha$ is given for a range of: a2) basal friction correlation lengths, b2) basal friction variances and c2) step sizes in basal friction reduction. The Dorfberg power law exponent and its fit uncertainty are indicated in gray.

The snow density random field, which also determines the bond strengths, showed little influence on the release area distribution (Figure 7). The remaining input parameters (snow height, slope angle, $s_{strength}$, residual friction, Figure 8) had limited effect on the maximum release area sizes, but the power law exponent was influenced by the slope angle and residual friction $\mu_{res}$. The power law exponent reached a minimum around a slope angle of 35° and increasing residual friction led to an increase in the power law exponent (Figure 8).

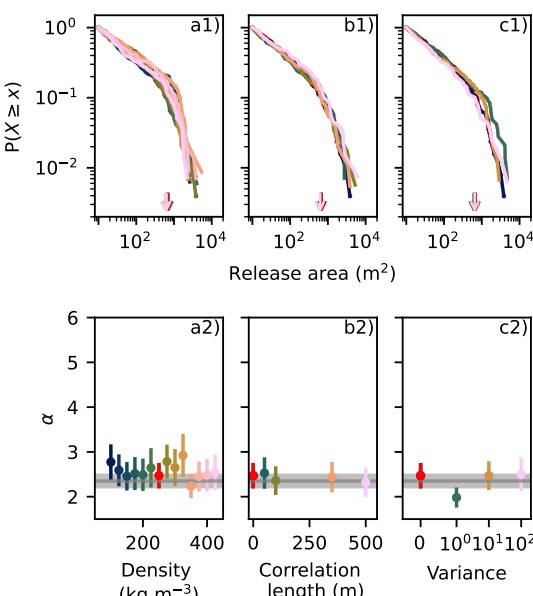

**Figure 7.** Effect of the snow density on the release area distribution for a range of: a1) densities, b1) density correlation lengths, and c1) density variances. The color indicates the parameter value as given in the graph below. The arrows indicate $x_{\min}$ which was set at 633 m$^2$ for all simulations. The power law exponent $\alpha$ is given for a range of: a2) densities, b2) density correlation lengths and c2) density variances. The Dorfberg power law exponent and its fit uncertainty are indicated in gray.

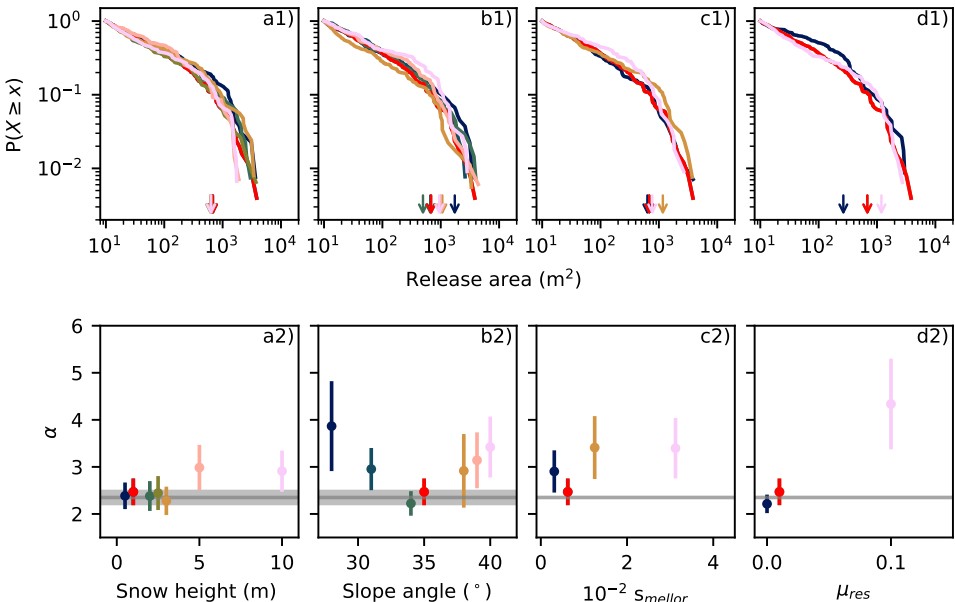

**Figure 8.** Effect of the a1) snow height, b1) slope angle, c1) scaling factor $s_{\text{strength}}$, and c1) residual friction $\mu_{\text{res}}$ on the release area distribution. The color indicates the parameter value as given in the graph below. The arrows indicate $x_{\text{min}}$. The power law exponent $\alpha$ is given for a range of a2) snow heights, b2) slope angles, c2) scaling factors and d2) residual frictions. The Dorfberg power law exponent and its fit uncertainty are indicated in gray.

### 4.3 Real topography

The power law exponent of avalanches that released on the Seewer Berg slope ($\alpha_{\text{Seewer Berg}} = 2.1 \pm 0.2$, $x_{\text{min}}$ = 273 m$^2$) was comparable to the power law exponent that we obtained running the model on the DEM of the corresponding slope ($\alpha = 2.2 \pm 0.2$, $x_{\text{min}}$ = 10 m$^2$, Figure 9a). The aspect ratio of the release areas on the DEM ($\frac{\text{width}}{\text{length}} = 1.0 \pm 0.3$) increased slightly compared to the uniform slope simulations ($\frac{\text{width}}{\text{length}} = 0.8 \pm 0.3$). The qualitative comparison of the release area locations in the model (Figure 9b) with the heatmap created from field observations (Figure 9c) showed an overlap in the most likely release locations. However, the model underestimated the large release area sizes. This resulted in an overall offset of the release area distribution to smaller release areas.

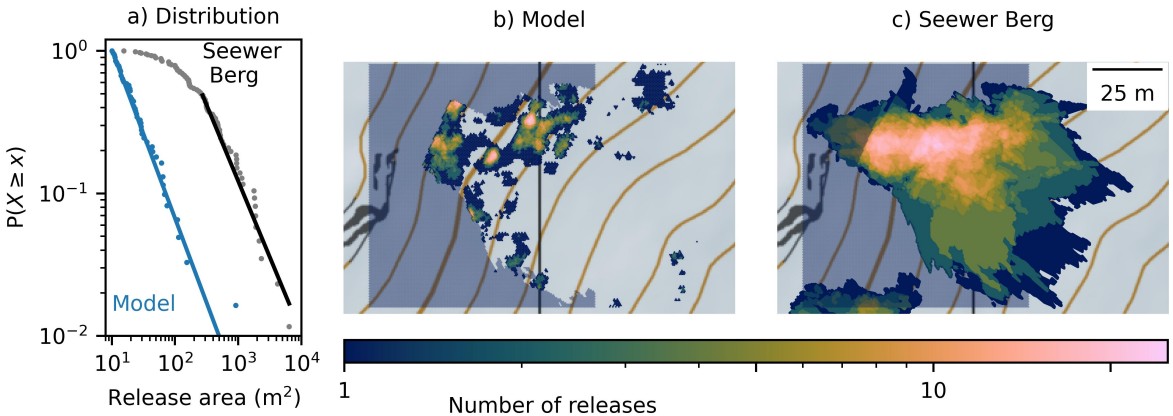

**Figure 9.** a) Release area distribution of recorded glide snow avalanches on the Seewer Berg slope (black, $\alpha_{\text{Seewer Berg}} = 2.1 \pm 0.2$, $x_{\text{min}} = 273\,\text{m}^2$) and the simulation based on the DEM (blue, $\alpha = 2.2 \pm 0.2$, $x_{\text{min}} = 10\,\text{m}^2$) with the corresponding power law fit (line plot). b) The heatmap of modelled avalanche release with the masked out area due to the high vegetation roughness (blue shaded). c) The heatmap of observed glide-snow avalanche releases (2008/09 to 2022/23) with the masked out area (blue shaded). Map: Federal Office of Topography

## 5 Discussion

### Observed glide-snow avalanche release area distribution

We observed glide-snow avalanche release areas on Dorfberg for 15 seasons. The release area distribution can be described with a power law distribution with an exponent of $\alpha_{\text{Dorfberg}} = 2.4 \pm 0.1$ for release areas larger than 633 m². To the best of our knowledge, there are no comparable datasets for glide-snow avalanches available. However, a similar exponent has been found for dry-snow slab avalanches ($\alpha_{\text{slab}} = 2.2 \pm 0.1$, Faillettaz et al. (2004)). Our findings are limited to release areas at Dorfberg and local topographic properties likely impact the observed distribution. In addition, the extraction of release areas from time-

lapse photographs is inherently limited in resolution. The minimum detectable release area depends on the topography and the orientation of the release area towards the camera (Fees et al., 2023). As a result, the number of very small avalanches may be underrepresented in the Dorfberg dataset.

The power law distribution was a suitable fit for the observed glide-snow avalanche release area distribution on Dorfberg (Figure 5a). The Kolmogorov-Smirnov-Test yielded p = 0.88 which supports the null hypothesis that the data follow

a power law distribution (Clauset et al., 2009). However, the log normal distribution could be another suitable distribution ($p_{\text{log normal}} = 0.99$). An increased number of observed glide-snow avalanche release areas across several orders of magnitudes and across varying locations would be necessary to confirm scale-invariant power law behavior.

Large field datasets may also allow for the separation of avalanches based on the suspected source of interfacial water (surface-generated interfacial water vs. interface-generated interfacial water (Fees et al., 2023)). For the Dorfberg observa-

240 tions, the release area distribution of interface events ($\alpha_{\text{interface}} = (4.5 \pm 0.9)$, $x_{\text{min}} = 1649\,\text{m}^2$, n = 16) exhibited a larger exponent than for surface events ($\alpha_{\text{surface}} = (2.1 \pm 0.2)$, $x_{\text{min}} = 640\,\text{m}^2$, n = 41). However, this result is considered preliminary

due to scarce observation data.

## Model assumptions and limitations

Based on the observation that glide-snow avalanche release areas on Dorfberg can be described with a power law, we built a threshold-based model of many interacting snow columns. The assumptions in the model are based on our current understanding of the processes leading to glide-snow avalanche release. There are two model assumptions that should to be discussed. (i) We assumed that the basal friction decreases uniformly across the entire slope. This assumption may hold well for avalanches in spring when melt water percolation can cause wetting across the entire slope. For avalanches in early winter, this assumption may be less appropriate. We assume that the interfacial water in early winter originates from geothermal heat melting the basal snow or from capillary suction of water from the soil into the snowpack (McClung, 1987; Mitterer and Schweizer, 2012). Both processes would allow for local increases in interfacial water due to, for example, locally higher soil temperature or soil saturation (Lombardo et al., 2023). These processes may be better represented in the model with a locally decreasing basal friction. (ii) We assumed that the ratio between compressive, shear and tensile bond strengths is constant and the same as for dry snow (Mellor, 1975). This assumption may hold well for avalanches in early winter when the snowpack is predominantly dry, but less so in spring when the snowpack is wet. We kept this ratio constant for all simulations because it was the only implemented parametrization based on snow experiments. In a future development step, we may vary the ratio to analyze if it has on influence on the aspect ratio of the glide-snow avalanche release area.

As of today, the main limitation in our understanding of glide-snow avalanches is the lack of knowledge on wet-snow mechanics and the formation/influence of liquid water. There are only a few measurements on the mechanical properties of wet snow available (Yamanoi and Endo, 2002; Izumi and Akitaya, 1985; Schlumpf et al., 2024). In the model, this prevents the parameterization and introduction of dependencies between more model parameters related to snow. The basal friction, snow density, and snow bond strengths could potentially all be connected to the liquid water content at the ground-snow interface. Linking more parameters to each other is an important step towards driving the model with a physical quantity such as the snow liquid water content.

## Model results

Although the model is built on numerous assumptions and simplifications, it reproduces the power law distribution of glide-snow avalanche release areas observed on Dorfberg (Figure 5). The spatial variation in basal friction and the friction step size were the dominant parameters for determining the power law exponent of the release area distribution and also had a substantial influence on the maximum release area size. This suggests that the uniformity of basal friction, as well as whether it transitions gradually (with a small friction step size) or abruptly (with a large friction step size), impacts the distribution of release areas for glide-snow avalanches. This observation is in line with the findings of the stauchwall model (Bartelt et al., 2012) which pointed out the importance of the length of the gliding zone. An investigation using SOC concepts on the basal friction reduction of a hanging glacier also found that the area and rate of the decreasing friction influenced instability (Faillettaz et al., 2011). For glide-snow avalanches the influence of basal friction uniformity on avalanche release has to be verified through field measure-

ments. In fact, spatio-temporal soil liquid water content measurements in the Seewer Berg slope (season 2021/22 and 2023/24) showed an increase in spatial uniformity (decreasing variance and/or increasing correlation length) before avalanche release (Fees et al., in preparation). Variations in snowpack properties (density, bond strengths) and the snow height had little influence
on the power law exponent and the maximum release area size. This is in line with our observations on Dorfberg which showed a weak correlation between snow height and glide-snow avalanche release area (r=0.22, p<0.001, Pearson correlation). The slope angle and residual friction did not influence the maximum release size, but did affect the power law exponent of the distribution. This may indicate the importance of the slope angle and surface roughness (which could be linked to the residual friction) for glide-snow avalanche release. In the future, the influence of the slope angle and surface roughness can be investi-
gated with the model in more detail on the Seewer Berg topography. A proxy for the surface roughness (e.g. vector ruggedness measure (Sappington et al., 2007)) could be extracted from summer drone orthophotos and implemented as the residual friction.

**Model limitations on topography**

The model run on a real topography showed that the model power law exponent was comparable to the field observations in
the Seewer Berg slope. In contrast to the modeled distribution on a uniform slope (Figure 5), the modeled distribution on the topography suggested power law behaviour across all magnitudes of simulated release areas. We suppose that the local slope angles dominated the location of avalanche release and that the boundary conditions introduced by the system size were not as constraining as on the uniform slope. Increasing the system size on the uniform slope also resulted in a release area distribution which suggests power law behaviour at smaller release areas (Figure A1c).
The location of simulated glide-snow avalanches qualitatively matched the typical release locations from field observations. This indicates that the topography has a substantial influence on the location of avalanche release (in line with Lackinger (1987); Leitinger et al. (2008); Peitzsch et al. (2015)). The aspect ratio of the simulated release areas on the topography is marginally larger than on the uniform slope and thus closer to the Dorfberg field observations. This may indicate that the topography has an influence on the aspect ratio. However, the simulation systematically underestimates the total release area.
This was also the case when we increased the hexagon cross section. It would be possible to rescale the release areas in a post processing step without changing the power law exponent. However, to more accurately simulate the release area size, further sensitivity analysis and/or the implementation of more parameter relationships are necessary.

**Model potential**

The promising reproduction of the Dorfberg glide-snow avalanche release area distribution with the model illustrates the potential of using a non-linear model for glide-snow avalanches. In the stauchwall model (Bartelt et al., 2012) the likelihood of avalanche release depends strongly on the length of the gliding zone which is currently unknown. The pseudo-3D setup of our model has the potential to narrow down typical length-scales for the gliding zone also depending on the topography. In the future, a more rigorous implementation of the snow cover, its mechanical properties, and the interaction with the soil would be
necessary. However, this is currently limited by the availability of data and parameterizations linking snow density, liquid water

content, and basal friction. Linking more parameters could help improve the underestimation of release areas on a complex topography.

The statistical nature of the model enables the investigation of different hypotheses to improve our understanding of parameter combinations that lead to critical glide-snow avalanche release conditions (high probability for large release areas). The model accounts for spatial variability (e.g. snow cover or surface roughness) which can help narrow down the length scales at which time-intensive (snow) observations have to be conducted for more targeted investigations.

## 6 Conclusions

We presented a mechanical, threshold-based model for the release area distribution of glide-snow avalanches. It was based on our current understanding of glide-snow avalanche release. The model consists of many interacting snow columns on a uniform slope that are driven towards a critical state through a reduction in basal friction. The snow column interaction with its neighbors can result in a chain-reaction of failing columns leading to avalanche release.

This model was able to reproduce the power law exponent of the release area distribution which we observed on Dorfberg during 15 seasons (n = 488). The sensitivity analysis of the model input parameters showed that the variability (variance and correlation length) of the basal friction as well as the gradual (small step size) or sudden (large step size) basal friction reduction had a substantial influence on the power law exponent. Snow cover related parameters (density correlation length, variance, snow bond strengths) had less influence on the power law exponent. Expanding the model onto the real topography extracted from a digital elevation model showed that the location of simulated release areas qualitatively matched the locations observed in the field, suggesting that the topography is important for the location of glide-snow avalanche release.

In the future, the model can grow in complexity with our growing understanding of the processes causing glide-snow avalanche release. Input parameters such as the basal friction, snow density, and snow bond strengths could potentially all be connected to the liquid water content at the ground-snow interface to drive the model with the liquid water content or meteorological measurements. The model has the potential to help identify potentially dangerous conditions for large or numerous avalanches, which would improve glide-snow avalanche forecasting.

## Appendix A: Boundary conditions

The boundary conditions (Table 1) were kept fixed for all simulations. Here we motivate our choice of boundary conditions and discuss their potential influence on the release area distribution.

The number of simulations (Figure A1a) was set at 30 simulations. This number of simulations resulted in a number of avalanches comparable to the number of avalanches observed on Dorfberg (n≈500). An increase in the number of simulations (n=100) did not substantially influence the power law exponent (Figure A1a2) but increased computation time.

The spatial dependencies of the basal friction random field were modelled with an exponential covariance function. An exponential covariance function has been used to describe spatial structure of soil water content (Delbari et al., 2009; Korres et al., 2015). In addition it was able to qualitatively improve the model distribution at small release areas in comparison to a Gaussian covariance function (Figure A1b1).

In order to simulate a power law distribution without a $x_{\min}$-cutoff an infinite system size would be needed (Amitrano, 2012).

Our model has a finite system size which can limit the occurrence of large release areas and result in a power law distribution that is affected by an exponential tail (Amitrano, 2012). We observed that with increasing system size (1500 x 1500 hexagons), the cutoff decreased ($x_{\min} = 10$ m$^2$) and the similarity to a theoretical power law distribution increased (Figure A1c1). We did not find an indication that the distribution of large release areas or the maximum release area was influenced by the system size. This may indicate that the correlation length of the random field is more constraining than the system size for large release

areas. As a compromise between system size and simulation duration we set a system size of 100 x 100 hexagons.

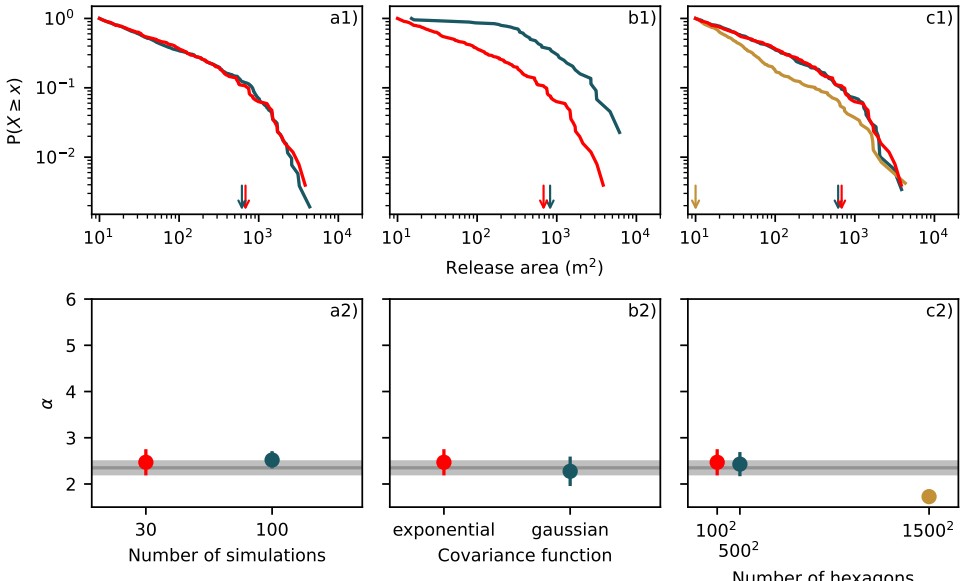

**Figure A1.** _1) Effect of the boundary conditions on the release area distribution. For the 1500 x 1500 system size only 15 simulations were performed due to long simulation duration. The color indicates the parameter value as given in the graph below. The arrows indicate $x_{\min}$. _2) Comparison between the modelled power law exponents $\alpha$ and the Dorfberg power law exponent and its fit uncertainty (gray).

*Code and data availability.* The data and code is available by request.

*Author contributions.* Conceptualization: all authors, Methodology: A.F., P.L., Software: A.F., Formal analysis: A.F., Data curation: A.F., A.H, Visualization: A.F., Writing - original draft: A.F., Writing review & editing: all authors, Supervision: A.H., P.L., J.S, Funding acquisition: P.L., J.S.

*Competing interests.* The authors have no competing interests to declare.

*Acknowledgements.* This research was supported by the Swiss National Science Foundation (grant no. 200021-212949). Colormaps by Crameri et al. (2020). The authors would like to thank Grégoire Bobillier for helpful discussions, Erich Peitzsch as editor and the reviewers Jérome Faillettaz and Christoph Mitterer for helpful and in-depth comments.

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
