# Peer review of "Glide-snow avalanches: A mechanical, threshold-based release area model"

_Natural Hazards and Earth System Sciences, 2024_

## Author Comment (AC1)

We would like to thank Jérome Faillettaz for his helpful and in-depth review of our manuscript.

This paper investigates the mechanism of glide-snow avalanche through a threshold-based model. Despite its simplicity, such models have yielded compelling results by aiming to replicate the emergent behavior—specifically, the release of avalanches—by employing basic interacting elements. This approach seeks to minimize the number of parameters while capturing the statistical behavior of complex phenomena. Consequently, it allows for the examination of the relative impact of chosen parameters on the overall emergence of phenomena and enhances the qualitative understanding of the phenomenon under investigation.

Having already demonstrated success in modeling landslides, this threshold-based model is now extended to the domain of glide-snow avalanche release. Leveraging data from a specific field site, this study enables the comparison of numerical and field results and the testing of various hypotheses. The findings highlight the significance of heterogeneity and the evolution of basal friction properties in the dynamics of avalanche release. This study introduces a new framework that underscores the primary influence of friction on the triggering mechanisms of snow avalanches.

Furthermore, the initial attempt to apply this model to a real slope with realistic parameters shows promise and hints at further fruitful research.

This paper addresses relevant scientific questions with an original approach. The paper is well-written, logically organized, clear, and well-structured. I believe this excellent work deserves to be published in NHESS after clarifying some points:

**General comments:**

My primary concern revolves around the justification of the power law behavior observed in glide avalanche release areas. The entire framework of the paper is built upon the assumption that these release areas follow a power law distribution, serving as the fundamental basis for the overall approach and numerical model. However, it's worth noting that the field data utilized in the study were collected solely from a single site, which may not provide conclusive evidence of such behavior. While I am inclined to support this assumption, the authors should exercise caution in making such assertions. Although they acknowledge in the discussion section the necessity for additional data from different field sites to validate this behavior, it's crucial to emphasize this limitation. Universality class might change for different slopes, aspect, slope orientation…

Furthermore, while I am convinced of the relevance of employing Self-Organized Criticality (SOC) concepts to model avalanche release, I still have some doubts regarding the numerical findings. Specifically, the detection of power law behavior solely at the extreme tail of the distribution raises concerns, as it primarily affects only a few large avalanches over less than one order of magnitude. The authors mention the evaluation of the power law exponent using the maximum likelihood method and $x_{min}$ using Clauset's method. However, to enhance the rigor of the analysis, it would be worth to compare different candidate distributions, such as

lognormal and power law, and provide the p-value for a more comprehensive assessment. By doing so, this study would achieve a greater degree of robustness and credibility.

Thank you for this valuable feedback. In addition to the power law, we also fitted a positive log-normal distribution to the release area distribution of Dorfberg and the baseline simulation (Figure 1). The p-values (Kolmogorov-Smirnov-Test, Figure 1) show that both, the power law and log normal distribution are good candidate distributions for the Dorfberg observations.

To adress this point, we will add a section to the discussion where we discuss the current data limitations. We will also formulate statements around the power law distribution on Dorfberg more cautiously throughout the manuscript.

[Figure]

Figure 1: Comparison of a power law and log normal fit for the Dorfberg and model baseline release area distribution. The p-values are: $p_{\text{power law\_Dorfberg}} = 0.88$, $p_{\text{log normal\_Dorfberg}} = 0.99$, $p_{\text{power law\_Model}} = 0.52$, $p_{\text{log normal\_Model}} = 0.93$ which support the null hypothesis that the observations/simulations follow the power law or log normal distribution.

It could be worthwhile to mention in the introduction that other types of models, which address the interplay between sliding, friction, and tension cracking using the concepts of SOC, exist. These include spring-block model types such as the Olami–Feder–Christensen model, Burridge Knopoff model, and others. Additionally, various other models exist to study the fracture process, such as the Random Fuse Model, Fiber Bundle Model, percolation (Alava et al., 2006)…

We will add a section to the introduction to point out other types of SOC models and their applications to mass movements or snow failure.

**Specific comments:**

Line 44: I would suggest a more cautious formulation: "These heavy-tailed power law distributions may potentially be associated with SOC."

We will implement this change as suggested.

L.46: Other models utilizing these concepts can replicate such behavior, including the spring-block model (e.g., Burridge-Knopoff type), fiber bundle models, thermal fuse model, branching model, among others (Sornette, 2006). For statistical fracture models, please also refer to Alava et al., 2006.

As mentioned above, we will add a section to the introduction to point out other types of models and their applications to mass movements or snow failure. We will also cite the work of Alava et al. (2006), thanks for this suggestion.

L.70: Using a Gaussian random field with an exponential covariance function seems reasonable for reproducing the spatial fluctuations of the friction coefficient on a slope. Is there any evidence of such variation in spatial properties in nature? Could you provide any references? How does the initial distribution of friction affect your results? Would the results differ if the friction coefficient were initialized with a uniform random distribution?

We used an exponential covariance function because we observed that it qualitatively improved the modeled release area distribution for small release area compared to a Gaussian covariance function (Figure 2b). In addition, we monitored the Seewer Berg slope with a grid consisting of 24 soil liquid water content sensors (spacing around 8 m × 8 m) as a proxy for interfacial water which is suspected to be a main driver for basal friction reduction. The analysis of these data also showed that an exponential function was a better fit to describe the spatial relationship of these sensor measurements using a variogram (Figure 3, in preparation for publication).

It was shown in several studies that the spatial structure of soil water content can be described by an exponential function. Yates and Warrick (1987), Mello et al. (2011), and Yang et al. (2018) used both exponential and spherical models for soils in USA, Brazil and China, respectively and Delhari et al. (2009) for a case study from Austria and Korres et al. (2015) for a study in Germany found that the exponential variogram shows the best performance.

To adress this and later comments on the influence of model parameters on the release area distribution we will add a section in the Appendix including Figure 2. In this section we will also refer to the available studies on the spatial structure of the soil water content.

[Figure]

Figure 2: _1) Release area distribution for different boundary conditions of the model – a) the number of simulations, b) the covariance function used in the random field, and c) the number of hexagons in the simulation domain compared to the baseline model (red). _2) The power law exponent α in comparison to the Dorfberg exponent and fit uncertainty (gray). The error bars indicate the fit uncertainty.

[Figure]

Figure 3: Example of the soil volumetric water content data in the Seewer Berg slope (23 April 2024) in form of a semi-variogram (uniform binning, exponential and Gaussian covariance function). Within the correlation length of about 40 meters, the semi-variogram can be well described with an exponential function.

Section 2.2: I'm not entirely certain about how bonds are handled. From what I understand, during the inspection of each cell, the failure of each bond is evaluated in shear, tension, and compression, and stress is redistributed according to those that remain intact. Is there any memory of bonds? In other words, if a bond between (q,r) and (q+1,r) failed in shear during the inspection of (q,r), will this failure be taken into account when inspecting (q+1,r)?

In the current version of the model no memory of bonds is implemented. If  the bond between (q,r) and (q+1, r) fails but (q,r) can be stabilized by the remaining bonds this does not influence the evaluation of (q+1,r). Only if (q,r) fails the bond is removed which influences the stability evaluation of (q+1,r). The implementation of memory in the bonds would be an interesting future addition to the model. We will clarify this in the manuscript.

Section 2.5: The explanation of the weighting factor was clear. I was just wondering how the authors deal with the case where gamma = 0. Do they consider only one compressive bond, or three? Do they arbitrarily select two bonds among the three to be in compression?

Thanks for this insightful comment. The case of gamma = 0 is currently not specifically implemented. The downslope directions of our topography did not exhibit this extreme case of alignment with one hexagon.

Table 1: Why are there so few simulations, only 30? How long does a typical run last?

We chose 30 simulations because, on average, this resulted in a number of simulated avalanches in the order of magnitude comparable to observations at Dorfberg. The aim was to keep the modeled and observed distribution comparable. We did an examplary study on the baseline simulation to determine if the number of simulations influences the power law exponent (Figure 2a). We found that more simulations did not influence α substantially. We will address this in the boundary conditions section in the Appendix and Figure 2a.

There was also a tradeoff between the simulation run time and the (extreme) input parameter combinations. For the baseline simulation parameters, a simulation run typically lasts ~30 seconds. For simulation runs with generally more stable conditions, many stable iterations were needed and the simualtion run time varied substantially also upwards of 30 minutes. Our priority was to keep the number of simulations constant throughout the sensitivity analaysis. In the future there is a lot of potential in speeding up the simulation run time through optimizing the number of iterations needed and more (computationally) efficient identification of stable state conditions.

 L.185: Figure 5b appears to depict a specific run where the outcome shows only one avalanche with an area of 848 m2 and four others of 3 m2 (which are not counted). Obtaining more than 500 avalanches with only 30 runs seems improbable based on this representation. It's possible that the figure is illustrating a particular case or scenario rather than a typical run. Further clarification from the authors may be necessary to reconcile this observation with the reported results of more than 500 avalanches from the 30 runs.

We will point out in the manuscript that this is an extreme case with a large release area. Other simulation runs result in many small avalanche releases which contribute substantially to the overall number of avalanche releases.

In Figure 8, the distributions look very similar despite the variation in alpha values from 2 to 5…

We will rescale the plot (Figure 4) for improved visualization.

[Figure]

Figure 4: Rescaled version of Figure 8 in the original manuscript.

The emergence of a pure power-law (i.e. without cut-off) is theoretically possible only within a system of infinite size. In the case of a finite size system, the occurrence of the largest events is constrained by the size of the system. As a consequence, the power-law distribution is affected by an exponential tail (Amitrano, 2012). At first glance, the distributions displayed in Figures 6, 7, and 8 appear to follow a power-law distribution with some cutoff, possibly related to finite size effects. Have you attempted simulations with a higher number of cells? ($10^4$ cells may not be sufficient to capture the full range of behavior).

We performed simulations also with dimensions of $500 \times 500$ and $1500 \times 1500$ hexagons to investigate finite size effects. We did not observe substantially larger release areas or a change in the occurence of these events. We suspect this may be due to the correlation length of the underlying random field dominating the maximum release area more substantially than the system size. However, we observed that with larger system size the simulated distribution is closer to a power law distribution at smaller release areas (in line with the theoretical observations of Amitrano 2012). We will address this in the boundary conditions section of the Appendix (Figure 2c).

Figure 9a illustrates an almost power-law distributed release area for the model, even for small avalanches, a difference from the results observed in the baseline model. The authors may ponder the underlying reasons for this discrepancy. Could the mask used in the simulations have influenced this outcome? Alternatively, might it be due to the varying slope angles across the lattice in these simulations? Including a brief discussion of these factors in the discussion section would be beneficial.

We suspect that the local slope angles dominate the location of avalanche release and that the boundary conditions introduced by the system size are not as constraining as on the uniform slope. Increasing the system size on the uniform slope also resulted in a release area distribution which suggests power law behaviour at smaller release areas (Figure 2c).

We will add this to the discussion.

The discussion section is quite interesting and raises important points. During my review of this paper, I noted a few additional remarks:

1. Does the aspect ratio depend on the relative sharing magnitude (f) of 10:2:1? I suspect that higher shear would enlarge the avalanche ratio.

   This is a very interesting point for further model development and analysis. The relative sharing magnitude was one of the very few parameterizations that is based on experimental snow data. In order to reduce the number of free variables we did not specifically investigate the relationship of the ratio on the aspect ratio in this manuscript. We will add this to the discussion.

2. The results presented in this paper are quite similar to those of Faillettaz et al. (2011). Although that study focused on instabilities in hanging glaciers, a similar investigation involving changes in friction coefficient was conducted, and similar effects were observed.

   Thank you for pointing this similarity out. We will include this in the discussion where we put the simulation results in context with the stauchwall model by Bartelt et al. (2012) as suggested by the second reviewer.

3. Why not consider water basal discharge, such as drainage paths (computed with slope map), as a proxy for friction decrease? In this way, friction would decrease preferentially along flow paths (e.g., gullies), in relationship with intensity of melting.

   Thank you for this suggestion. Drainage paths or a similar index (e.g. the terrain wetness index) could be a good candidate to further quantify the basal friction. Our vision for the future was to also quantify the basal friction by combining the vegetation roughness (quantified from drone orthophotos) and the grid of soil liquid water content sensors which we installed in the Seewer Berg slope. These sensors provided spatio-temporal soil liquid water content measurements before and during avalanche release (Figure 3, in preparation for publication).

References:

Bartelt, P., Feistl, T., Bühler, Y., and Buser, O.: Overcoming the stauchwall: Viscoelastic stress redistribution and the start of full-depth gliding snow avalanches, Geophys. Res. Lett., 39, 2012GL052479, https://doi.org/10.1029/2012GL052479, 2012.

Yates, S. R., & Warrick, A. W. (1987). Estimating soil water content using cokriging. Soil Science Society of America Journal, 51(1), 23-30.

Delbari, M., Afrasiab, P., & Loiskandl, W. (2009). Using sequential Gaussian simulation to assess the field-scale spatial uncertainty of soil water content. Catena, 79(2), 163-169.

Korres, W., Reichenau, T. G., Fiener, P., Koyama, C. N., Bogena, H. R., Cornelissen, T., ... & Schneider, K. (2015). Spatio-temporal soil moisture patterns–A meta-analysis using plot to catchment scale data. Journal of hydrology, 520, 326-341.

Mello, C. R. D., Ávila, L. F., Norton, L. D., Silva, A. M. D., Mello, J. M. D., & Beskow, S. (2011). Spatial distribution of top soil water content in an experimental catchment of Southeast Brazil. Scientia Agricola, 68, 285-294.

Yang, Y., Huang, Y., Zhang, Y., & Tong, X. (2018). Optimal irrigation mode and spatio-temporal variability characteristics of soil moisture content in different growth stages of winter wheat. Water, 10(9), 1180.

Reference

Alava, M.J., Nukala, P.K.V.V., Zapperi, S., 2006. Statistical models of fracture. AP 55, 349–476. https://doi.org/10.1080/00018730300741518

Amitrano, D., 2012. Variability in the power-law distributions of rupture events. The European Physical Journal Special Topics 205, 199–215. https://doi.org/10.1140/epjst/e2012-01571-9

Faillettaz, J., Sornette, D., Funk, M., 2011. Numerical modeling of a gravity-driven instability of a cold hanging glacier: reanalysis of the 1895 break-off of Altelsgletscher, Switzerland. JG 57, 817–831. https://doi.org/10.3189/002214311798043852

---

## Author Comment (AC2)

**Summary**

The authors present to my knowledge one of the few mechanical-based model approaches to better understand the release behavior of glide-snow avalanches. The basic concept of the model approach relies on a self-organized criticality (SOC) approach, which has been applied to various other, similar rapid gravitational mass movements. Since our knowledge on release of glide-snow avalanches is notoriously limited, the authors are forced to make various assumptions and parameterizations. All of them are valid and plausible. The modelling approach is two-fold. First, the authors apply their model assumptions and governing equations to a simplified, uniform slope and perform a sensitivity analysis. Then, they apply the model to a complex topography and compare the results to observed data from the test site Dorfberg above Davos, Switzerland. The model results reveal interesting insights into the components that are relevant to the release of glide-snow avalanches, but more importantly sets the stage for further more detailed investigations using the presented approach.

**Evaluation**

The presented manuscript applies transparently a sound set of methods to obtain innovative results on the mechanical processes relevant to glide-snow avalanche release. Approach and results are scientifically relevant and represent a major impact on that specific topic for the community.

The manuscript is concise, well-structured and nicely written. The reader can easily follow the thoughts and approaches of the authors. Figures and tables are clearly structured and adequately described. I am convinced that this excellent work should be published on NHESS after addresses some additional points.

**General remarks**

In general, I miss a little more discussion and context to previous studies, especially on those that already pointed to some drivers that may be very important for the release of glide-snow avalanches. I think by adding some more details and discussion, the manuscript would highly gain impact especially on narrowing down some of the processes relevant for glide-snow avalanche release.

Here are some of my thoughts:

- You compare the model results to observed glide-snow avalanche data without stratifying according to surface vs. interface events (cp. Fees et al., 2023). Are there differences in the observed $x_{min}$ and alpha when accounting for the different events?

We investigated the power law exponent and $x_{min}$ separated into surface and interface events (Figure 1) and observed that interface events showed a larger power law exponent than surface events. However, as the other reviewer also pointed out, the available data for large avalanche events is currently very limited. We will point out the difference in the power law exponent between interface and surface events in the discussion as an indication for further research. We will also put this 'preliminary' finding in context of the currently limited data availability.

[Figure]

Figure 1: Dorfberg avalanche release area distribution separated in surface and interface events (points) and the corresponding power law fit (line). The arrows indicate $x_{min.}$

- While you nicely show that snow density and snow height do not have large impact on especially alpha, I was wondering if changes in snow height, e.g. amount of new snow would lead to more sensitive reactions of the model. This in turn would be interesting since especially interface events (formally also called cold-temperature events) seem to highly react on added mass of snow (Dreier et al., 2016).

At the moment the model is based on the assumption that a spatially uniform reduction in basal friction drives the instability of the slope. If we assume a model setup where the basal friction distribution does not decrease (but is in a potentially critical state) and uniform snow loading drives the model we would expect that the snow loading has a similar influence on the release area distribution as the 'basal friction reduction step size' (Manuscript Figure 6c). Both the reduction in basal friction and snow loading would be a spatially uniform contribution towards instability. To investigate the simultaneous basal friction decrease and snow loading the snow loading would have to be implemented in the model. This would be an interesting step in further model development.

- On the other side you show that variance and correlation length of the basal friction have large impact on the power law fits. Both results (not sensitive to snow height, sensitive to the correlation length of the basal friction) were already mentioned in Bartelt et al. (2012). I think the community would highly benefit, if you could discuss in more detail where your and the results by Bartelt et al. (2012) show similar and/or different behaviour and why and where assumptions of both models may have contributed to the agreement or differences.

Thanks for this insightful comment. We will refer to the stauchwall model in the discussion of the results to highlight similarities and differences. In the 'model potential' section we will point out that our model can help to narrow down the length scales of the gliding zone which is one of the major assumptions in the stauchwall model.

**Specific remarks**

- Lines 124-125: The term "stable state" might be a bit misleading here.

  We will clarify the definition of "stable state".

- Table 1: Why did you use 30 simulation runs and are the number of runs relevant for the results?

  We chose 30 simulation runs because, on average, this resulted in a number of simulated avalanches in the order of magnitude comparable to the Dorfberg field observations. The aim was to keep the modeled and observed distribution comparable. We did an exemplary study on the baseline simulation to determine if the number of simulations influences the power law exponent. We found that more simulations did not influence α substantially. We will address this in a boundary conditions section in the Appendix (Figure 1a, more details are provided in reply to the first reviewer).

- Line 231-232: Can you underpin a little more your statement: "For avalanches in early winter ...".

  We will extend the discussion by including the following statements: "For avalanches in early winter, this assumption may be less appropriate. We suspect that the interfacial water in early winter originates from geothermal heat melting the snow at the bottom of the snow cover or from capillary suction of water from the soil into the snowpack. Both processes would allow for local increases in interfacial water due to, for example, locally higher soil temperature or soil saturation."

[Figure]

Figure 1: _1) Release area distribution for different boundary conditions of the model – a) the number of simulations, b) the covariance function used in the GRF, and c) the number of hexagons in the simulation domain compared to the baseline model (red). _2) The power law exponent α in comparison to the Dorfberg exponent and fit uncertainty (gray). The error bars indicate the fit uncertainty.

**Literature**

Bartelt, P., Feistl, T., Bühler, Y., and Buser, O.: Overcoming the stauchwall: Viscoelastic stress redistribution and the start of full-depth gliding snow avalanches, Geophys. Res. Lett., 39, 2012GL052479, https://doi.org/10.1029/2012GL052479, 2012.

Dreier, L., Harvey, S., van Herwijnen, A., and Mitterer, C.: Relating meteorological parameters to glide-snow avalanche activity, Cold Reg. Sci. Technol., 128, 57–68, https://doi.org/10.1016/j.coldregions.2016.05.003, 2016.

Fees, A., Van Herwijnen, A., Altenbach, M., Lombardo, M., and Schweizer, J.: Glide-snow avalanche characteristics at different timescales extracted from time-lapse photography, Ann. Glaciol., 1–12, https://doi.org/10.1017/aog.2023.37, 2023.